# Compression-aware Training of Deep Networks

**Jose M. Alvarez**
Toyota Research Institute
Los Altos, CA 94022
jose.alvarez@tri.global

**Mathieu Salzmann**
EPFL - CVLab
Lausanne, Switzerland
mathieu.salzmann@epfl.ch

## Abstract

In recent years, great progress has been made in a variety of application domains thanks to the development of increasingly deeper neural networks. Unfortunately, the huge number of units of these networks makes them expensive both computationally and memory-wise. To overcome this, exploiting the fact that deep networks are over-parametrized, several compression strategies have been proposed. These methods, however, typically start from a network that has been trained in a standard manner, without considering such a future compression. In this paper, we propose to explicitly account for compression in the training process. To this end, we introduce a regularizer that encourages the parameter matrix of each layer to have low rank during training. We show that accounting for compression during training allows us to learn much more compact, yet at least as effective, models than state-of-the-art compression techniques.

## 1 Introduction

With the increasing availability of large-scale datasets, recent years have witnessed a resurgence of interest for Deep Learning techniques. Impressive progress has been made in a variety of application domains, such as speech, natural language and image processing, thanks to the development of new learning strategies [15, 53, 30, 45, 26, 3] and of new architectures [31, 44, 46, 23]. In particular, these architectures tend to become ever deeper, with hundreds of layers, each of which containing hundreds or even thousands of units.

While it has been shown that training such very deep architectures was typically easier than smaller ones [24], it is also well-known that they are highly over-parameterized. In essence, this means that equally good results could in principle be obtained with more compact networks. Automatically deriving such equivalent, compact models would be highly beneficial in runtime- and memory-sensitive applications, e.g., to deploy deep networks on embedded systems with limited hardware resources. As a consequence, many methods have been proposed to compress existing architectures.

An early trend for such compression consisted of removing individual parameters [33, 22] or entire units [36, 29, 38] according to their influence on the output. Unfortunately, such an analysis of individual parameters or units quickly becomes intractable in the presence of very deep networks. Therefore, currently, one of the most popular compression approaches amounts to extracting low-rank approximations either of individual units [28] or of the parameter matrix/tensor of each layer [14]. This latter idea is particularly attractive, since, as opposed to the former one, it reduces the number of units in each layer. In essence, the above-mentioned techniques aim to compress a network that has been pre-trained. There is, however, no guarantee that the parameter matrices of such pre-trained networks truly have low-rank. Therefore, these methods typically truncate some of the relevant information, thus resulting in a loss of prediction accuracy, and, more importantly, do not necessarily achieve the best possible compression rates.

In this paper, we propose to explicitly account for compression while training the initial deep network. Specifically, we introduce a regularizer that encourages the parameter matrix of each layer to have

low rank in the training loss, and rely on a stochastic proximal gradient descent strategy to optimize the network parameters. In essence, and by contrast with methods that aim to learn uncorrelated units to prevent overfitting [5, 54, 40], we seek to learn correlated ones, which can then easily be pruned in a second phase. Our compression-aware training scheme therefore yields networks that are well adapted to the following post-processing stage. As a consequence, we achieve higher compression rates than the above-mentioned techniques at virtually no loss in prediction accuracy.

Our approach constitutes one of the very few attempts at explicitly training a compact network from scratch. In this context, the work of [4] has proposed to learn correlated units by making use of additional noise outputs. This strategy, however, is only guaranteed to have the desired effect for simple networks and has only been demonstrated on relatively shallow architectures. In the contemporary work [51], units are coordinated via a regularizer acting on all pairs of filters within a layer. While effective, exploiting all pairs can quickly become cumbersome in the presence of large numbers of units. Recently, group sparsity has also been employed to obtain compact networks [2, 50]. Such a regularizer, however, acts on individual units, without explicitly aiming to model their redundancies. Here, we show that accounting for interactions between the units within a layer allows us to obtain more compact networks. Furthermore, using such a group sparsity prior in conjunction with our compression-aware strategy lets us achieve even higher compression rates.

We demonstrate the benefits of our approach on several deep architectures, including the 8-layers DecomposeMe network of [1] and the 50-layers ResNet of [23]. Our experiments on ImageNet and ICDAR show that we can achieve compression rates of more than 90%, thus hugely reducing the number of required operations at inference time.

## 2   Related Work

It is well-known that deep neural networks are over-parametrized [13]. While, given sufficient training data, this seems to facilitate the training procedure, it also has two potential drawbacks. First, over-parametrized networks can easily suffer from overfitting. Second, even when they can be trained successfully, the resulting networks are expensive both computationally and memory-wise, thus making their deployment on platforms with limited hardware resources, such as embedded systems, challenging. Over the years, much effort has been made to overcome these two drawbacks.

In particular, much progress has been made to reduce overfitting, for example by devising new optimization strategies, such as DropOut [45] or MaxOut [16]. In this context, other works have advocated the use of different normalization strategies, such as Batch Normalization [26], Weight Normalization [42] and Layer Normalization [3]. Recently, there has also been a surge of methods aiming to regularize the network parameters by making the different units in each layer less correlated. This has been achieved by designing new activation functions [5], by explicitly considering the pairwise correlations of the units [54, 37, 40] or of the activations [9, 52], or by constraining the weight matrices of each layer to be orthonormal [21].

In this paper, we are more directly interested in addressing the second drawback, that is, the large memory and runtime required by very deep networks. To tackle this, most existing research has focused on pruning pre-trained networks. In this context, early works have proposed to analyze the saliency of individual parameters [33, 22] or units [36, 29, 38, 34], so as to measure their impact on the output. Such a local analysis, however, quickly becomes impractically expensive when dealing with networks with millions of parameters.

As a consequence, recent works have proposed to focus on more global methods, which analyze larger groups of parameters simultaneously. In this context, the most popular trend consists of extracting low-rank approximations of the network parameters. In particular, it has been shown that individual units can be replaced by rank 1 approximations, either via a post-processing step [28, 46] or directly during training [1, 25]. Furthermore, low-rank approximations of the complete parameter matrix/tensor of each layer were computed in [14], which has the benefit of reducing the number of units in each layer. The resulting low-rank representation can then be fine-tuned [32], or potentially even learned from scratch [47], given the rank of each layer in the network. With the exception of this last work, which assumes that the ranks are known, these methods, however, aim to approximate a given pre-trained model. In practice, however, the parameter matrices of this model might not have low rank. Therefore, the resulting approximations yield some loss of accuracy and, more importantly,

will typically not correspond to the most compact networks. Here, we propose to explicitly learn a low-rank network from scratch, but without having to manually define the rank of each layer a priori.

To this end, and in contrast with the above-mentioned methods that aim to minimize correlations, we rather seek to maximize correlations between the different units within each layer, such that many of these units can be removed in a post-processing stage. In [4], additional noise outputs were introduced in a network to similarly learn correlated filters. This strategy, however, is only justified for simple networks and was only demonstrated on relatively shallow architectures. The contemporary work [51] introduced a penalty during training to learn correlated units. This, however, was achieved by explicitly computing all pairwise correlations, which quickly becomes cumbersome in very deep networks with wide layers. By contrast, our approach makes use of a low-rank regularizer that can effectively be optimized by proximal stochastic gradient descent.

Our approach belongs to the relatively small group of methods that explicitly aim to learn a compact network during training, i.e., not as a post-processing step. Other methods have proposed to make use of sparsity-inducing techniques to cancel out individual parameters [49, 10, 20, 19, 35] or units [2, 50, 55]. These methods, however, act, at best, on individual units, without considering the relationships between multiple units in the same layer. Variational inference [17] has also been used to explicitly compress the network. However, the priors and posteriors used in these approaches will typically zero out individual weights. Our experiments demonstrate that accounting for the interactions between multiple units allows us to obtain more compact networks.

Another line of research aims to quantize the weights of deep networks [48, 12, 18]. Note that, in a sense, this research direction is orthogonal to ours, since one could still further quantize our compact networks. Furthermore, with the recent progress in efficient hardware handling floating-point operations, we believe that there is also high value in designing non-quantized compact networks.

## 3 Compression-aware Training of Deep Networks

In this section, we introduce our approach to explicitly encouraging compactness while training a deep neural network. To this end, we propose to make use of a low-rank regularizer on the parameter matrix in each layer, which inherently aims to maximize the compression rate when computing a low-rank approximation in a post-processing stage. In the following, we focus on convolutional neural networks, because the popular visual recognition models tend to rely less and less on fully-connected layers, and, more importantly, the inference time of such models is dominated by the convolutions in the first few layers. Note, however, that our approach still applies to fully-connected layers.

To introduce our approach, let us first consider the $l$-th layer of a convolutional network, and denote its parameters by $\theta_l \in \mathbb{R}^{K_l \times C_l \times d_l^H \times d_l^W}$, where $C_l$ and $K_l$ are the number of input and output channels, respectively, and $d_l^H$ and $d_l^W$ are the height and width of each convolutional kernel. Alternatively, these parameters can be represented by a matrix $\hat{\theta}_l \in \mathbb{R}^{K_l \times S_l}$ with $S_l = C_l d_l^H d_l^W$. Following [14], a network can be compacted via a post-processing step performing a singular value decomposition of $\hat{\theta}_l$ and truncating the 0, or small, singular values. In essence, after this step, the parameter matrix can be approximated as $\hat{\theta}_l \approx U_l M_l^T$, where $U_l$ is a $K_l \times r_l$ matrix representing the basis kernels, with $r_l \leq \min(K_l, S_l)$, and $M_l$ is an $S_l \times r_l$ matrix that mixes the activations of these basis kernels.

By making use of a post-processing step on a network trained in the usual way, however, there is no guarantee that, during training, many singular values have become near-zero. Here, we aim to explicitly account for this post-processing step during training, by seeking to obtain a parameter matrix such that $r_l << \min(K_l, S_l)$. To this end, given $N$ training input-output pairs $(\mathbf{x}_i, y_i)$, we formulate learning as the regularized minimization problem

$$\min_{\Theta} \frac{1}{N} \sum_{i=1}^{N} \ell(y_i, f(\mathbf{x}_i, \Theta)) + r(\Theta) , \tag{1}$$

where $\Theta$ encompasses all network parameters, $\ell(\cdot, \cdot)$ is a supervised loss, such as the cross-entropy, and $r(\cdot)$ is a regularizer encouraging the parameter matrix in each layer to have low rank.

Since explicitly minimizing the rank of a matrix is NP-hard, following the matrix completion literature [7, 6], we make use of a convex relaxation in the form of the nuclear norm. This lets us

write our regularizer as

$$r(\Theta) = \tau \sum_{l=1}^{L} \|\hat{\theta}_l\|_* ,\qquad(2)$$

where $\tau$ is a hyper-parameter setting the influence of the regularizer, and the nuclear norm is defined as $\|\hat{\theta}_l\|_* = \sum_{j=1}^{rank(\hat{\theta}_l)} \sigma_l^j$, with $\sigma_l^j$ the singular values of $\hat{\theta}_l$.

In practice, to minimize (1), we make use of proximal stochastic gradient descent. Specifically, this amounts to minimizing the supervised loss only for one epoch, with learning rate $\rho$, and then applying the proximity operator of our regularizer. In our case, this can be achieved independently for each layer. For layer $l$, this proximity operator corresponds to solving

$$\theta_l^* = \underset{\bar{\theta}_l}{\text{argmin}} \frac{1}{2\rho} \|\bar{\theta}_l - \hat{\theta}_l\|_F^2 + \tau \|\bar{\theta}_l\|_* ,\qquad(3)$$

where $\hat{\theta}_l$ is the current estimate of the parameter matrix for layer $l$. As shown in [6], the solution to this problem can be obtained by soft-thresholding the singular values of $\hat{\theta}_l$, which can be written as

$$\theta_l^* = U_l \Sigma_l(\rho\tau) V_l^T, \quad \text{where } \Sigma_l(\rho\tau) = diag([(\sigma_l^1 - \rho\tau)_+, \dots, (\sigma_l^{rank(\hat{\theta}_l)} - \rho\tau)_+]),\qquad(4)$$

$U_l$ and $V_l$ are the left - and right-singular vectors of $\hat{\theta}_l$, and $(\cdot)_+$ corresponds to taking the maximum between the argument and 0.

## 3.1 Low-rank and Group-sparse Layers

While, as shown in our experiments, the low-rank solution discussed above significantly reduces the number of parameters in the network, it does not affect the original number of input and output channels $C_l$ and $K_l$. By contrast, the group-sparsity based methods [2, 50] discussed in Section 2 cancel out entire units, thus reducing these numbers, but do not consider the interactions between multiple units in the same layer, and would therefore typically not benefit from a post-processing step such as the one of [14]. Here, we propose to make the best of both worlds to obtain low-rank parameter matrices, some of whose units have explicitly been removed.

To this end, we combine the sparse group Lasso regularizer used in [2] with the low-rank one described above. This lets us re-define the regularizer in (1) as

$$r(\Theta) = \sum_{l=1}^{L} \left( (1-\alpha)\lambda_l \sqrt{P_l} \sum_{n=1}^{K_l} \|\theta_l^n\|_2 + \alpha\lambda_l \|\theta_l\|_1 \right) + \tau \sum_{l=1}^{L} \|\hat{\theta}_l\|_* ,\qquad(5)$$

where $K_l$ is the number of units in layer $l$, $\theta_l^n$ denotes the vector of parameters for unit $n$ in layer $l$, $P_l$ is the size of this vector (the same for all units in a layer), $\alpha \in [0,1]$ balances the influence of sparsity terms on groups vs. individual parameters, and $\lambda_l$ is a layer-wise hyper-parameter. In practice, following [2], we use only two different values of $\lambda_l$; one for the first few layers and one for the remaining ones.

To learn our model with this new regularizer consisting of two main terms, we make use of the incremental proximal descent approach proposed in [39], which has the benefit of having a lower memory footprint than parallel proximal methods. The proximity operator for the sparse group Lasso regularizer also has a closed form solution derived in [43] and provided in [2].

## 3.2 Benefits at Inference

Once our model is trained, we can obtain a compact network for faster and more memory-efficient inference by making use of a post-processing step. In particular, to account for the low rank of the parameter matrix of each layer, we make use of the SVD-based approach of [14]. Specifically, for each layer $l$, we compute the SVD of the parameter matrix as $\hat{\theta}_l = \tilde{U}_l \tilde{\Sigma}_l \tilde{V}_l$ and only keep the $r_l$ singular values that are either non-zero, thus incurring no loss, or larger than a pre-defined threshold, at some potential loss. The parameter matrix can then be represented as $\hat{\theta}_l = U_l M_l$, with $U_l \in \mathbb{R}^{C_l d_l^H d_l^W \times r_l}$ and $M_l = \Sigma_l V_l \in \mathbb{R}^{r_l \times K_l}$. In essence, every layer is decomposed into two layers. This incurs significant memory and computational savings if $r_l(C_l d_l^H d_l^W + K_l) << (C_l d_l^H d_l^W K_l)$.

Furthermore, additional savings can be achieved when using the sparse group Lasso regularizer discussed in Section 3.1. Indeed, in this case, the zeroed-out units can explicitly be removed, thus yielding only $\hat{K}_l$ filters, with $\hat{K}_l < K_l$. Note that, except for the first layer, units have also been removed from the previous layer, thus reducing $C_l$ to a lower $\hat{C}_l$. Furthermore, thanks to our low-rank regularizer, the remaining, non-zero, units will form a parameter matrix that still has low rank, and can thus also be decomposed. This results in a total of $r_l(\hat{C}_l d_l^H d_l^W + \hat{K}_l)$ parameters.

In our experiments, we select the rank $r_l$ based on the percentage $e_l$ of the energy (i.e., the sum of singular values) that we seek to capture by our low-rank approximation. This percentage plays an important role in the trade-off between runtime/memory savings and drop of prediction accuracy. In our experiments, we use the same percentage for all layers.

## 4 Experimental Settings

**Datasets:** For our experiments, we used two image classification datasets: ImageNet [41] and ICDAR, the character recognition dataset introduced in [27]. ImageNet is a large-scale dataset comprising over 15 million labeled images split into $22,000$ categories. We used the ILSVRC-2012 [41] subset consisting of 1000 categories, with 1.2 million training images and $50,000$ validation images. The ICDAR dataset consists of 185,639 training samples combining real and synthetic characters and 5,198 test samples coming from the ICDAR2003 training set after removing all non-alphanumeric characters. The images in ICDAR are split into 36 categories. The use of ICDAR here was motivated by the fact that it is fairly large-scale, but, in contrast with ImageNet, existing architectures haven't been heavily tuned to this data. As such, one can expect our approach consisting of training a compact network from scratch to be even more effective on this dataset.

**Network Architectures:** In our experiments, we make use of architectures where each kernel in the convolutional layers has been decomposed into two 1D kernels [1], thus inherently having rank-1 kernels. Note that this is orthogonal to the purpose of our low-rank regularizer, since, here, we essentially aim at reducing the number of kernels, not the rank of individual kernels. The decomposed layers yield even more compact architectures that require a lower computational cost for training and testing while maintaining or even improving classification accuracy. In the following, a convolutional layer refers to a layer with 1D kernels, while a decomposed layer refers to a block of two convolutional layers using 1D vertical and horizontal kernels, respectively, with a non-linearity and batch normalization after each convolution.

Let us consider a decomposed layer consisting of $C$ and $K$ input and output channels, respectively. Let $\bar{v}$ and $\bar{h}^T$ be vectors of length $d^v$ and $d^h$, respectively, representing the kernel size of each 1D feature map. In this paper, we set $d^h = d^v \equiv d$. Furthermore, let $\varphi(\cdot)$ be a non-linearity, and $x_c$ denote the $c$-th input channel of the layer. In this setting, the activation of the $i$-th output channel $f_i$ can be written as

$$f_i = \varphi(b_i^h + \sum_{l=1}^{L} \bar{h}_{il}^T * [\varphi(b_l^v + \sum_{c=1}^{C} \bar{v}_{lc} * x_c)]), \tag{6}$$

where $L$ is the number of vertical filters, corresponding to the number of input channels for the horizontal filters, and $b_l^v$ and $b_l^h$ are biases.

We report results with two different models using such decomposed layers: DecomposeMe [1] and ResNets [23]. In all cases, we make use of batch-normalization after each convolutional layer [1]. We rely on rectified linear units (ReLU) [31] as non-linearities, although some initial experiments suggest that slightly better performance can be obtained with exponential linear units [8]. For DecomposeMe, we used two different $\text{Dec}_8$ architectures, whose specific number of units are provided in Table 1. For residual networks, we used a decomposed ResNet-50, and empirically verified that the use of 1D kernels instead of the standard ones had no significant impact on classification accuracy.

**Implementation details:** For the comparison to be fair, all models, including the baselines, were trained from scratch on the same computer using the same random seed and the same framework. More specifically, we used the torch-7 multi-gpu framework [11].

| | 1v | 1h | 2v | 2h | 3v | 3h | 4v | 4h | 5v | 5h | 6v | 6h | 7v | 7h | 8v | 8h |
|---|---|---|---|---|---|---|---|---|---|---|---|---|---|---|---|---|
| $Dec_8^{256}$ | 32/11 | 64/11 | 128/5 | 192/5 | 256/3 | 384/3 | 256/3 | 256/3 | 256/3 | 256/3 | 256/3 | 256/3 | 256/3 | 256/3 | 256/3 | 256/3 |
| $Dec_8^{512}$ | 32/11 | 64/11 | 128/5 | 192/5 | 256/3 | 384/3 | 256/3 | 256/3 | 512/3 | 512/3 | 512/3 | 512/3 | 512/3 | 512/3 | 512/3 | 512/3 |
| $Dec_3^{512}$ | 48/9 | 96/9 | 160/9 | 256/9 | 512/8 | 512/8 | – | – | – | – | – | – | – | – | – | – |

Table 1: **Different DecomposeMe architectures used on ImageNet and ICDAR.** Each entry represents the number of filters and their dimension.

| Layer / Conf$_\lambda$ | 0 | 1 | 2 | 3 | 4 | 5 | 6 | 7 | 8 | 9 | 10 | 11 |
|---|---|---|---|---|---|---|---|---|---|---|---|---|
| 1v to 2h | – | 0.0127 | 0.051 | 0.204 | 0.255 | 0.357 | 0.051 | 0.051 | 0.051 | 0.051 | 0.051 | 0.153 |
| 3v to 8h | – | 0.0127 | 0.051 | 0.204 | 0.255 | 0.357 | 0.357 | 0.408 | 0.510 | 0.255 | 0.765 | 0.51 |

Table 2: **Sparse group Lasso hyper-parameter configurations.** The first row provides $\lambda$ for the first four convolutional layers, while the second one shows $\lambda$ for the remaining layers. The first five configurations correspond to using the same regularization penalty for all the layers, while the later ones define weaker penalties on the first two layers, as suggested in [2].

For ImageNet, training was done on a DGX-1 node using two-P100 GPUs in parallel. We used stochastic gradient descent with a momentum of 0.9 and a batch size of 180 images. The models were trained using an initial learning rate of 0.1 multiplied by 0.1 every 20 iterations for the small models ($Dec_8^{256}$ in Table 1) and every 30 iterations for the larger models ($Dec_8^{512}$ in Table 1). For ICDAR, we trained each network on a single TitanX-Pascal GPU for a total of 55 epochs with a batch size of 256 and 1,000 iterations per epoch. We follow the same experimental setting as in [2]: The initial learning rate was set to an initial value of 0.1 and multiplied by 0.1. We used a momentum of 0.9.

For DecomposeMe networks, we only performed basic data augmentation consisting of using random crops and random horizontal flips with probability 0.5. At test time, we used a single central crop. For ResNets, we used the standard data augmentation advocated for in [23]. In practice, in all models, we also included weight decay with a penalty strength of $1e^{-4}$ in our loss function. We observed empirically that adding this weight decay prevents the weights to overly grow between every two computations of the proximity operator.

In terms of hyper-parameters, for our low-rank regularizer, we considered four values: $\tau \in \{0, 1, 5, 10\}$. For the sparse group Lasso term, we initially set the same $\lambda$ to every layer to analyze the effect of combining both types of regularization. Then, in a second experiment, we followed the experimental set-up proposed in [2], where the first two decomposed layers have a lower penalty. In addition, we set $\alpha = 0.2$ to favor promoting sparsity at group level rather than at parameter level. The sparse group Lasso hyper-parameter values are summarized in Table 2.

**Computational cost:** While a convenient measure of computational cost is the forward time, this measure is highly hardware-dependent. Nowadays, hardware is heavily optimized for current architectures and does not necessarily reflect the concept of any-time-computation. Therefore, we focus on analyzing the number of multiply-accumulate operations (MAC). Let a convolution be defined as $f_i = \varphi(b_i + \sum_{j=1}^{C} W_{ij} * x_j)$, where each $W_{ij}$ is a 2D kernel of dimensions $d^H \times d^W$ and $i \in [1, \dots K]$. Considering a naive convolution algorithm, the number of MACs for a convolutional layer is equal to $PCKd^hd^W$ where $P$ is the number of pixels in the output feature map. Therefore, it is important to reduce $CK$ whenever $P$ is large. That is, reducing the number of units in the first convolutional layers has more impact than in the later ones.

## 5  Experimental Results

**Parameter sensitivity and comparison to other methods on ImagNet:** We first analyze the effect of our low-rank regularizer on its own and jointly with the sparse group Lasso one on MACs and accuracy. To this end, we make use of the $Dec_8^{256}$ model on ImageNet, and measure the impact of varying both $\tau$ and $\lambda$ in Eq. 5. Note that using $\tau = \lambda = 0$ corresponds to the standard model, and $\tau = 0$ and $\lambda \neq 0$ to the method of [2]. Below, we report results obtained without and with the post-processing step described in Section 3.2. Note that applying such a post-processing on the standard model corresponds to the compression technique of [14]. Fig. 1 summarizes the results of this analysis.

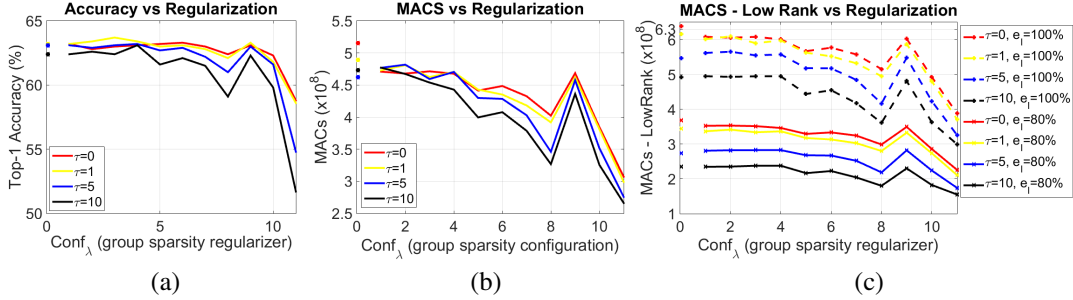

(a)                  (b)                  (c)

Figure 1: **Parameter sensitivity for $Dec_8^{256}$ on ImageNet**. **(a)** Accuracy as a function of the regularization strength. **(b)** MACs directly after training. **(c)** MACS after the post-processing step of Section 3.2 for $e_l = \{100\%, 80\%\}$. In all the figures, isolated points represent the models trained without sparse group Lasso regularizer. The red point corresponds to the baseline, where no low-rank or sparsity regularization was applied. The specific sparse group Lasso hyper-parameters for each configuration $Conf_\lambda$ are given in Table 2.

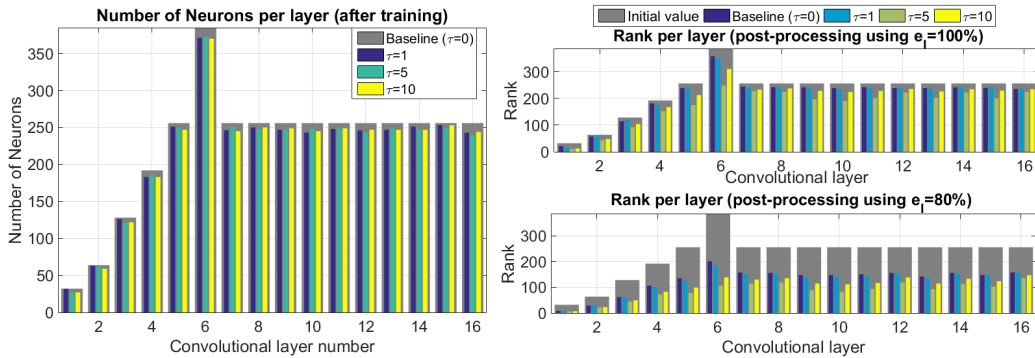

Figure 2: **Effect of the low-rank regularizer on its own on $Dec_8^{256}$ on ImageNet. (Left)** Number of units per layer. **(Right)** Effective rank per layer for (top) $e_l$=100% and (bottom) $e_l$=80%. Note that, on its own, our low-rank regularizer already helps cancel out entire units, thus inherently performing model selection.

In Fig. 1(a), we can observe that accuracy remains stable for a wide range of values of $\tau$ and $\lambda$. In fact, there are even small improvements in accuracy when a moderate regularization is applied.

Figs. 1(b,c) depict the MACs without and with applying the post-processing step discussed in Section 3.2. As expected, the MACs decrease as the weights of the regularizers increase. Importantly, however, Figs. 1(a,b) show that several models can achieve a high compression rate at virtually no loss in accuracy. In Fig. 1(c), we provide the curves after post-processing with two different energy percentages $e_l = \{100\%, 80\%\}$. Keeping all the energy tends to incur an increase in MAC, since the inequality defined in Section 3.2 is then not satisfied anymore. Recall, however, that, without post-processing, the resulting models are still more compact than and as accurate as the baseline one. With $e_l = 80\%$, while a small drop in accuracy typically occurs, the gain in MAC is significantly larger. Altogether, these experiments show that, by providing more compact models, our regularizer lets us consistently reduce the computational cost over the baseline.

Interestingly, by looking at the case where $Conf_\lambda = 0$ in Fig. 1(b), we can see that we already significantly reduce the number of operations when using our low-rank regularizer only, even without post-processing. This is due to the fact that, even in this case, a significant number of units are automatically zeroed-out. Empirically, we observed that, for moderate values of $\tau$, the number of zeroed-out singular values corresponds to complete units going to zero. This can be observed in Fig. 2(left), were we show the number of non-zero units for each layer. In Fig. 2(right), we further show the effective rank of each layer before and after post-processing.

| | Baseline | [14] | [2] | Ours | Ours+[2] | Ours+[2] |
|---|---|---|---|---|---|---|
| hyper-params | – | $e_l = 90\%$ | – | $\tau = 15, e_l = 90\%$ | $\tau = 15, e_l = 90\%$ | $\tau = 15, e_l = 100\%$ |
| # Params | 3.7M | 3.6M | 525K | 728K | 318K | 454K |
| top-1 | 88.6% | 88.5% | 89.6% | 88.8% | 89.7% | 90.5% |

Table 3: **Comparison to other methods on ICDAR.**

| | | Low-Rank appx. | | no SVD | | | | Low-Rank appx. | | no SVD | |
|---|---|---|---|---|---|---|---|---|---|---|---|
| Imagenet | Top-1 | Params | MAC | Params | MAC | ICDAR | Top-1 | Params | MAC | Params | MAC |
| $\text{Dec}_8^{512}$-$e_l$=80% | 66.8 | -53.5 | -46.2 | -39.5 | -25.3 | $\text{Dec}_3^{512}$-$e_l$=80% | 89.6 | -91.9 | -92.9 | -89.2 | -81.6 |
| $\text{Dec}_8^{512}$-$e_l$=100% | 67.6 | -21.1 | -4.8 | -39.5 | -25.3 | $\text{Dec}_3^{512}$-$e_l$=100% | 90.8 | -85.3 | -86.8 | -89.2 | -81.6 |

Table 4: **Accuracy and compression rates for $\text{Dec}_8^{512}$ models on ImageNet (left) and $\text{Dec}_3^{512}$ on ICDAR (right).** The number of parameters and MACs are given in % relative to the baseline model (i.e., without any regularizer). A negative value indicates reduction with respect to the baseline. The accuracy of the baseline is 67.0 for ImageNet and 89.3 for ICDAR.

| | $e_l = 80\%$ | $e_l = 100\%$ | no SVD | baseline ($\tau = 0$) |
|---|---|---|---|---|
| $\text{Dec}_8^{256}$-$\tau = 1$ | 97.33 | 125.44 | 94.60 | 94.70 |
| $\text{Dec}_8^{256}$-$\tau = 5$ | 88.33 | 119.27 | 90.55 | 94.70 |
| $\text{Dec}_8^{256}$-$\tau = 10$ | 85.78 | 110.35 | 91.36 | 94.70 |

Table 5: **Forward time in milliseconds (ms) using a Titan X (Pascal).** We report the average over 50 forward passes using a batch size of 256. A large batch size minimizes the effect of memory overheads due to non-hardware optimizations.

**Comparison to other approaches on ICDAR:** We now compare our results with existing approaches on the ICDAR dataset. As a baseline, we consider the $\text{Dec}_3^{512}$ trained using SGD and L2 regularization for 75 epochs. For comparison, we consider the post-processing approach in [14] with $e_l = 90\%$, the group-sparsity regularization approach proposed in [2] and three different instances of our model. First, using $\tau = 15$, no group-sparsity and $e_l = 90\%$. Then, two instances combining our low-rank regularizer with group-sparsity (Section 3.1) with $e_l = 90\%$ and $e_l = 100\%$. In this case, the models are trained for 55 epochs and then reloaded and fine tuned for 20 more epochs. Table 3 summarizes these results. The comparison with [14] clearly evidences the benefits of our compression-aware training strategy. Furthermore, these results show the benefits of further combining our low-rank regularizer with the groups-sparsity one of [2].

In addition, we also compare our approach with L1 and L2 regularizers on the same dataset and with the same experimental setup. Pruning the weights of the baseline models with a threshold of $1e - 4$ resulted in 1.5M zeroed-out parameters for the L2 regularizer and 2.8M zeroed-out parameters for the L1 regularizer. However, these zeroed out weights are sparsely located within units (neurons). Applying our post-processing step (low-rank approximation with $e_l = 100\%$) to these results yielded models with 3.6M and 3.2M parameters for L2 and L1 regularizers, respectively. The top-1 accuracy for these two models after post-processing was 87% and 89%, respectively. Using a stronger L1 regularizer resulted in lower top-1 accuracy. By comparison, our approach yields a model with 3.4M zeroed-out parameters after post-processing and a top-1 accuracy of 90%. Empirically, we found the benefits of our approach to hold for varying regularizer weights.

**Results with larger models:** In Table 4, we provide the accuracies and MACs for our approach and the baseline on ImageNet and ICDAR for $\text{Dec}_8^{512}$ models. Note that using our low-rank regularizer yields more compact networks than the baselines for similar or higher accuracies. In particular, for ImageNet, we achieve reductions in parameter number of more than 20% and more than 50% for $e_l = 100\%$ and $e_l = 80\%$, respectively. For ICDAR, these reductions are around 90% in both cases.

We now focus on our results with a ResNet-50 model on ImageNet. For post-processing we used $e_l = 90\%$ for all these experiments which resulted in virtually no loss of accuracy. The baseline corresponds to a top-1 accuracy of 74.7% and 18M parameters. Applying the post-processing step on this baseline resulted in a compression rate of 4%. By contrast, our approach with low-rank yields a top-1 accuracy of 75.0% for a compression rate of 20.6%, and with group sparsity and low-rank

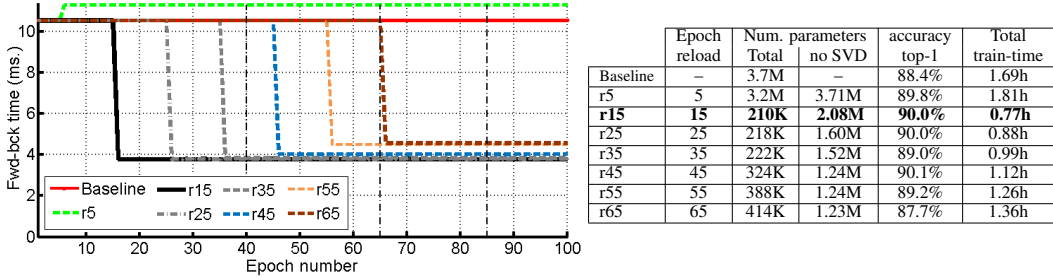

Figure 3: **Forward-Backward training time in milliseconds when varying the reload epoch for Dec$_3^{512}$ on ICDAR. (Left)** Forward-backward time per batch in milliseconds (with a batch size of 32). **(Right)** Summary of the results of each experiment. Note that we could reduce the training time from 1.69 hours (baseline) to 0.77 hours by reloading the model at the 15th epoch. This corresponds to a relative training-time speed up of 54.5% and yields a 2% improvement in top-1 accuracy.

jointly, a top-1 accuracy of 75.2% for a compression rate of 27%. By comparison, applying [2] to the same model yields an accuracy of 74.5% for a compression rate of 17%.

**Inference time:** While MACs represent the number of operations, we are also interested in the inference time of the resulting models. Table 5 summarizes several representative inference times for different instances of our experiments. Interestingly, there is a significant reduction in inference time when we only remove the zeroed-out neurons from the model. This is a direct consequence of the pruning effect, especially in the first layers. However, there is no significant reduction in inference time when post-processing our model via a low-rank decomposition. The main reason for this is that modern hardware is designed to compute convolutions with much fewer operations than a naive algorithm. Furthermore, the actual computational cost depends not only on the number of floating point operations but also on the memory bandwidth. In modern architectures, decomposing a convolutional layer into a convolution and a matrix multiplication involves (with current hardware) additional intermediate computations, as one cannot reuse convolutional kernels. Nevertheless, we believe that our approach remains beneficial for embedded systems using customized hardware, such as FPGAs.

**Additional benefits at training time:** So far, our experiments have demonstrated the effectiveness of our approach at test time. Empirically, we found that our approach is also beneficial for training, by pruning the network after only a few epochs (e.g., 15) and reloading and training the pruned network, which becomes much more efficient. Specifically, Table 3 summarizes the effect of varying the reload epoch for a model relying on both low-rank and group-sparsity. We were able to reduce the training time (with a batch size of 32 and training for 100 epochs) from 1.69 to 0.77 hours (relative speedup of 54.5%). The accuracy also improved by 2% and the number of parameters reduced from 3.7M (baseline) to 210K (relative 94.3% reduction). We found this behavior to be stable across a wide range of regularization parameters. If we seek to maintain accuracy compared to the baseline, we found that we could achieve a compression rate of 95.5% (up to 96% for an accuracy drop of 0.5%), which corresponds to a training time reduced by up to 60%.

## 6   Conclusion

In this paper, we have proposed to explicitly account for a post-processing compression stage when training deep networks. To this end, we have introduced a regularizer in the training loss to encourage the parameter matrix of each layer to have low rank. We have further studied the case where this regularizer is combined with a sparsity-inducing one to achieve even higher compression. Our experiments have demonstrated that our approach can achieve higher compression rates than state-of-the-art methods, thus evidencing the benefits of taking compression into account during training. The SVD-based technique that motivated our approach is only one specific choice of compression strategy. In the future, we will therefore study how regularizers corresponding to other such compression mechanisms can be incorporated in our framework.

## Footnotes

[1] We empirically found the use of batch normalization after each convolutional layer to have more impact with our low-rank regularizer than with group sparsity or with no regularizer, in which cases the computational cost can be reduced by using a single batch normalization after each decomposed layer.

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
