[Reviews · NeurIPS 2017]

Reviewer 1



The authors present a regularization term that encourages weight matrices to be low rank during network training, effectively increasing the compression of the network, and making it possible to explicitly reduce the rank during post-processing, reducing the number of operations required for inference. Overall the paper seems like a good and well written paper on an interesting topic. I have some caveats however: firstly the authors do not mention variational inference, which also explicitly compresses the network (in the literal sense of reducing the number of bits in its description length) and can be used to prune away many weights after training - see 'Practical Variational Inference for Neural Networks' for details. More generally, almost any regularizer provides a form of implicit 'compression-aware training' (that's why they work - simpler models generalize better) and can often be used to prune networks post-hoc. For example a network trained with l1 or l2 regularization will generally end up with many weights very close to 0, which can be removed without greatly altering network performance. I think it's important to clarify this, especially since the authors use an l2 term in addition to their own regularizer during training. They also don't compare seem to compare how well previous low rank post processing works with and without their regulariser, or with other regularisers used in previous work. All of these caveats could be answered by providing more baseline results in the experimental section, demonstrating that training with this particular regulariser does indeed lead to a better accuracy / compression tradeoff than other approaches. In general I found the results a little hard to interpret, so may be missing something: the graph I wanted to see was a set of curves for accuracy vs compression ratio (either in terms of number of parameters or number of MACs) rather than accuracy against the strength of the regularisation term. On this graph it should be possible to explicitly compare your approach vs previous regularisers / compressors.

Reviewer 2



This paper proposes a low-rank regularizer for deep model compression during training. Overall, this paper is well-written, and the motivation is clear. However, here are some comments as follows. 1 The novelty is relatively limited, as the technical parts are strongly relevant to the previous works. 2 The experiments should be further improved. (1) Parameter sensitivity: From Fig 1, the performance of the proposed method (\tau is 1,\lambda is not 0) is similar to [2] (\tau is 0,\lambda is not 0). For other settings of \tau, the compression rate is improved while the accuracy is reduced. (2) Results on larger models: the comparison with [2] should be performed to show the effectiveness. Furthermore, it would be interesting to compare with other state-of-the-art compression approaches, such as [18].

Reviewer 3



The paper is interesting. The propose training neural networks with a cost that explicitly favors networks that are easier to compress by truncated SVD. They formulate this regularization as a cost on the nuclear norm of the weight matrices, which they enforce with the soft threshold of the singular values as the proximal operator after every epoch. I found the idea interesting, and the experimental sections I thought gave a nice breakdown of the results of their own experiments and the behavior of their proposed method, but I would have liked to see some more comparative results, i.e. the performance of their own network versus other compression techniques targeting the same number of parameters on the datasets, for instance. Overall good paper, interesting idea, good execution, but experiments somewhat lacking.